# PRETRAINING FOR LANGUAGE-CONDITIONED IMITATION WITH TRANSFORMERS

## ABSTRACT

We study reinforcement learning (RL) agents which can utilize language inputs and efficiently learn on downstream tasks. To investigate this, we propose a new multimodal benchmark – Text-Conditioned Frostbite – in which an agent must complete tasks specified by text instructions in the Atari Frostbite environment. We curate and release a dataset of 5M text-labelled transitions for training, and to encourage further research in this direction. On this benchmark, we evaluate Text Decision Transformer (TDT), a transformer directly operating on text, state, and action tokens, and find it improves upon baseline architectures. Furthermore, we evaluate the effect of pretraining, finding unsupervised pretraining can yield improved results in low-data settings.

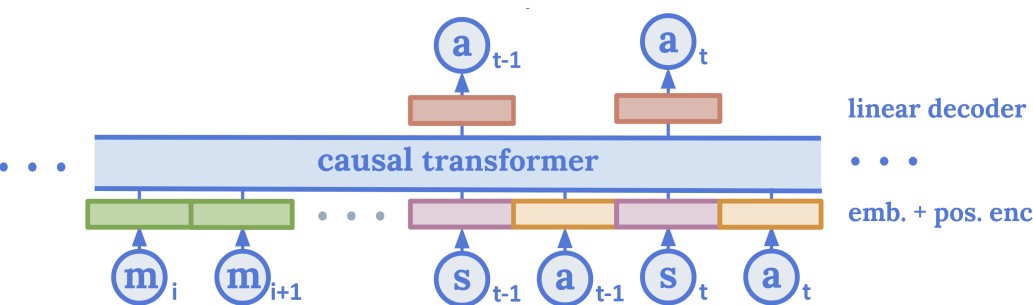

Figure 1: The Text-Conditioned Decision Transformer (TDT) architecture for specifying behaviors via language inputs. Text tokens $m_i$ are prepended to the sequence of episode tokens, specifying the current task. Alternating states $s_t$ and actions $a_t$ provide context for the model and are rolled out in the environment. Actions are predicted from every state in an autoregressive fashion.

## 1 INTRODUCTION

Whereas most contemporary reinforcement learning (RL) methods require programming a reward function, we are interested in training RL agents which understand language, enabling a flexible and interpretable interface and facilitating human-AI interaction. Natural language often carries tremendous amounts of information and can thus both provide strong learning signals and describe complex scenarios. Furthermore, agents which are able to learn from a diverse set of tasks – such as the space of tasks specified by language – can take advantage of broader data for supervision, access more diversity in an environment, and perform more effective few-shot learning. One path towards more general agents is *data-driven* reinforcement learning, which promises to leverage diverse data for improved generalization in RL (Levine et al., 2020), similar to approaches in natural language.

Recent approaches to data-driven RL have proposed leveraging powerful sequence models (Chen et al., 2021; Janner et al., 2021) that are now mainstream in language (Devlin et al., 2018; Radford et al., 2019). While initial approaches to training language models on large-scale unsupervised data focused on learning individual word embeddings (Pennington et al., 2014), newer models, especially those based on the transformer architecture (Vaswani et al., 2017), can also transfer sequential aspects learned during pretraining. Similar work in RL has found great benefit to transferring learned

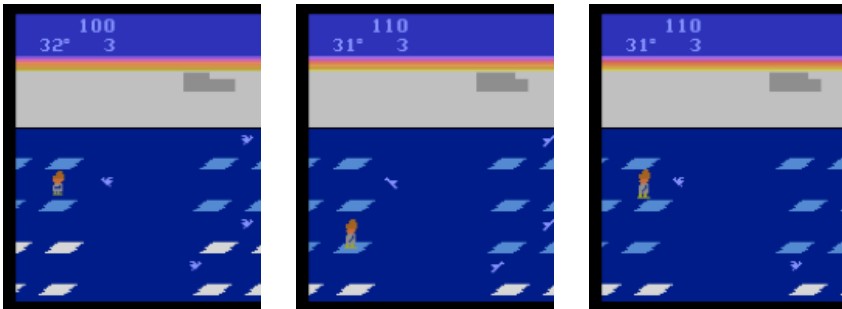

Task: "jump between the second and third ice floes"

Figure 2: We create a new dataset on the Atari Frostbite environment, consisting of 5M timesteps of labelled text-trajectory pairs. The agent must solve a diverse set of tasks specified by natural language instructions. Above are example frames representative of a task from the dataset.

embeddings via a vision encoder (Yarats et al., 2021), but transferring sequential behavior knowledge is relatively unstudied. However, using transformer sequence modeling approaches, we can develop methods which can finetune more effectively from offline data.

To better study this, we propose a new benchmark and release a new dataset[1], Text-Conditioned Frostbite, in which the agent must perform tasks specified by language instructions in the Atari Frostbite environment (Bellemare et al., 2013). Crucially, our dataset consists of a diverse set of tasks, ranging from easier tasks such as "try to stay on the right side" to harder tasks such as "spend as much time as possible on the fourth ice floe". Inspired by the proposal from Lake et al. (2017), we believe Frostbite is a diverse and dynamic environment requiring temporally extended strategies, whereas traditional evaluations underutilize the environment by finding agents with single, narrow strategies. We focus on Atari Frostbite as it allows for combining this naturalistic language with complex goals and behaviors using a familiar, well-studied environment.

We introduce a transformer operating on text-state-action sequences – which we call Text Decision Transformer (TDT) – adapting the single-stream Decision Transformer (Chen et al., 2021) to our setting of conditioning on language goals by replacing the reward tokens with text tokens. We perform empirical evaluations on our environment using the TDT, finding it serves as a reasonable baseline architecture for study, and evaluate performance with unsupervised pretraining.

Our contributions are as follows:

1. We propose a new multimodal benchmark, Text-Conditioned Frostbite, and provide a dataset of 5M labelled transitions with 500 different tasks.

2. We evaluate a single-stream architecture, Text Decision Transformer, and find it outperforms MLP baselines, highlighting sequence modeling-based approaches.

3. We investigate unsupervised pretraining, both on language and on trajectories, and find trajectory-based pretraining yields improved performance in low-data settings. Furthermore, we investigate scaling laws, finding the task is complex enough that TDT improves with more data.

## 2 PRELIMINARIES

### 2.1 LANGUAGE-CONDITIONED IMITATION LEARNING

In the problem of Language-Conditioned Imitation Learning, we are provided with a tuple of $(\mathcal{S}, \mathcal{A}, \mathcal{M}, \mathcal{P}, \mathcal{R})_i$, which correspond to the states, actions, tasks, transition dynamics, and a reward or score function. A trajectory $\tau = (s_1, a_1, ...s_t, a_t)$ is a sequence of states and actions generated by the transition dynamics and the agent's action at every timestep, where $t$ is less than or equal to the maximum episode length. The score function $\mathcal{R}$ is prompted with a trajectory $\tau$ and a text

---

[1]Dataset and code are available at: `<github will be made public at publication>`, temporary link to dataset and code for reproduction are in supplementary material

task $m$ and outputs the degree of success that the trajectory had with respect to the task – note we only use $\mathcal{R}$ for evaluation purposes. A policy/model $\pi_\theta(a_t|s_{\leq t}, a_{<t}, m)$ is tasked with outputting actions which maximize the score function, i.e. $\mathbb{E}_{a\sim\pi_\theta}[R(\tau, m)]$. The model has access to a dataset of expert trajectory-language pairs $(\tau, m)$ for training.

## 2.2 TRANSFORMERS

Transformers were originally proposed by Vaswani et al. (2017) and have since become the front-runner in solving sequence modelling problems. The models are built with sequential self-attention modules. Each self-attention module is given a sequence of $n$ embeddings and outputs another $n$ embeddings with the same dimensions. The self-attention modules work by performing the inner product between the current embedding $x_i$ with the query matrix $Q$ to give $q_i$ and with the key matrix $K$ to give the key $k_i$. The model takes the inner product between the key and query for $j \in [1, n]$, yielding $n$ logits, over which the model takes the softmax and yields a probability distribution. The final result is then a convex combination of the value vectors $v_i$, with the weights dictated by the probability distribution. More concisely, the $i$-th component of the output $z_i$ is given by

$$z_i = \sum_{j=1}^{n} \texttt{softmax}(\{\langle q_i, k_{j'}\rangle\}_{j'=1}^{n})_j \cdot v_j. \tag{1}$$

## 2.3 DECISION TRANSFORMER

Our work builds on Decision Transformer (DT) (Chen et al., 2021), which uses a GPT-2 (Radford et al., 2019) transformer architecture, leveraging a causal mask where the $i$-th prediction depends only on input tokens up to $i$. DT models a trajectory sequence $(\hat{R}_1, s_1, a_1, \ldots, \hat{R}_t, s_t, a_t)$ autoregressively, where $\hat{R}_t$ is the returns-to-go from timestep $t$ to the end of the episode. Test-time evaluation proceeds by initializing a behavior specification – DT uses a desired return $\hat{R}_1$ – and alternating generation between states from the environment and actions sampled from the model. To reduce computation, DT only uses the last $K$ timesteps as input to the model, called the context size. A context size of 1 represents a Markovian policy.

## 3 ENVIRONMENT AND DATASET

The first contribution of our work is in developing a diverse, yet accessible environment for evaluating the performance of text-conditioned imitation models. We build on the proposal from Lake et al. (2017), in which the authors propose the "Frostbite Challenge". The authors propose Frostbite is challenging due to the requirement of temporally extended strategies and the diverse landscape of possible tasks to complete. To construct a dataset for this task, we manually collected several hundred tasks and trajectories. These tasks range from "don't move" to "stay as long as you can on the fourth ice floe" to "die to a crab on your first life" and so on.

### 3.1 ENVIRONMENT DETAILS

**Evaluation via RAM state.** The first challenge with such an environment is building deterministic evaluation strategies. For this reason, evaluation of the agent is performed by accessing the RAM state of the game, calculating the level, position, score, and other related quantities of the agent, and comparing them with the desired outcomes. This evaluation loop runs after a trajectory is completed and can be prompted with a number of different desired tasks. The interface is malleable enough to even generate rewards for specific tasks, though we have not yet conducted such a study. We use the underlying state-space of the game to calculate specific metrics in the environment. These metrics are then normalized to be in the range $(0, 1)$, so no task gets unequal weighting. For example, in a task like "spend as much time as possible on the first ice floe", the assigned success is simply (number of timesteps on 1st ice floe) / (number of timesteps).

**Performance measurement.** Performance of the model is measured by prompting the model with several different tasks and subsequently running the evaluation procedure on the generated trajectories. The score is normalized to be in the range $[0, 1]$ and is designed to be a success metric. If the

| Split | Example Tasks |
|---|---|
| Easy | stay on the left side, die on the first level, get as close to 500 score as you can |
| Medium | jump between the second and third ice floes, reach level 2, die on the second level |
| Hard | reach level 5, spend as much time as possible on the fourth ice floe |

Table 1: Example tasks in our Text-Conditioned Frostbite dataset. Tasks are split into easy, medium, and hard categories for evaluation purposes.

model receives a score of 1 on a generated trajectory, then the model has succeeded in obeying the task, while a score of 0 indicates a failure.

## 3.2 DATASET DETAILS

**Overview.** The dataset consists of more than 700 labelled missions, for a total of 5m timesteps, each consisting of the current task, the current state, and the action taken. The Atari environment supports 18 discrete actions corresponding to directional movement and interaction. Game frames are resized and converted to black-white. The environment does not explicitly perform frame-stacking, as our agent has access to context, but frame-stacking can be implemented with the interface. We list some example tasks in Table 1 and include more in the Appendix.

**Collection.** We collected several hundred tasks and trajectories by hand. These trajectories were generated by taking advantage of the Atari-Gym user interface and playing Frostbite with keyboard controls. At the beginning of every task, users are prompted with a task that they will try to achieve, and their states, actions, and tasks are recorded and saved.

**Task categorizations.** For evaluation, a subset of the collected tasks are supported and separated into distinct task suites depending on the degree of control necessary to perform the task. For instance, in the easy task suite, there are tasks such as "dont move," "die on the first level," and "stay on the left side." Such tasks are quite simple and do not require advanced visual perception or control of the agent. In the medium task suite, tasks require higher levels of perception, but are less demanding than the hard suite. The medium task suite contains "get as close to 1000 score as you can," "reach level 2," and "jump between the second and third ice floes." The hard suite contains tasks that require the most advanced control, such as "reach level 5" and "spend as much time as possible on the fourth ice floe." The task suites include only tasks that support RAM score evaluation. A list of further training tasks can be found in the Appendix (see Table 3).

## 4 METHOD

Here, we detail the architecture and modifications of the Text-Conditioned Decision Transformer. This model uses a causal transformer architecture in the style of GPT-2 (Radford et al., 2019) to model sequences jointly consisting of text and trajectories.

## 4.1 ARCHITECTURE

Similar to Decision Transformer, we model the trajectories as a sequence of tokens. However, because we are operating within the framework of imitation learning, for behavior specification, we excise the return-to-go tokens from our architecture and use text tokens instead. These text tokens are prepended to the sequence of tokens. The original words of the task are encoded using an embedding table, whereas states and actions are encoded using CNNs / MLPs, respectively. Thus, a typical sequence of tokens passed into the model will resemble (see Figure 1):

$$\tau = (m_1, m_2, \ldots, m_n, s_1, a_1, s_2, a_2, \ldots, s_t, a_t)$$

The model is prompted with a text tokens and a sequence of states and actions. The inputs are tokenized and passed through the model. Action logits for action $a_t$ correspond to the output from state $s_t$, and due to the causal transformer mask, only tokens preceding $s_t$ are visible in this prediction. For generating trajectories, we sample from the predicted action logits, and step the environment

with the predicted action. At the end of the trajectory, we evaluate performance using the environment's score function to determine if the trajectory and original text are well-aligned.

We preprocess language inputs by converting to lower case and stripping any punctuation. Words are then converted to one-hot representation and passed through a learned embedding table. See Algorithm 1 for more details.

## 4.2 TRAINING

We operate with a standard supervised training procedure. Random minibatches of paired tasks and trajectories are sampled from our offline dataset. The model performs a forward pass over the joint sequence, outputting the logits of the distribution over next actions in parallel. We optimize the cross entropy loss between the predictions and the true actions. Pseudocode for the model and training is included in Algorithm 1.

## 4.3 UNSUPERVISED PRETRAINING

In Sections 5.2-5.4, we additionally evaluate the effect of unsupervised pretraining, which refers to training on unlabelled trajectories (without a text pair) drawn from the dataset. For pretraining, the model is only given as input states and actions, and otherwise follows the same procedure as the supervised training regime. The pretraining data is drawn from the corpus of all missions simply with text missions removed. Later, the model is fine-tuned and evaluated on specific subsets of missions to measure the effectiveness of the representations learned by pretraining. This is comparable to the standard autoregressive language modeling objective used by GPT (Radford et al., 2019), or behavior cloning in imitation learning. We can transfer the vision encoder layers and/or the sequence model (the transformer backbone).

## 5 EMPIRICAL EVALUATIONS

### 5.1 DOES THE TRANSFORMER ARCHITECTURE MODEL THE JOINT SEQUENCES WELL?

We first evaluate the transformer architecture itself in the fully supervised setting. We compare to an MLP baseline which removes the ability to attend between modalities: text tokens are processed by a pretrained BERT model (Devlin et al., 2018), and then concatenated to the state and action tokens, which are processed by the same encoders as TDT. The MLP with context (K) of 1 is chosen to have the same number of parameters as TDT. The MLP with context 1 is outperformed by all but one TDT architecture choice, even those with limited context. The MLP with context 20 is outperformed by the TDT architectures with context size of at least 10. This is despite the fact that the MLP with context size 20 uses many more parameters (since it relies on concatenating over timesteps). The TDT with one context represents a fairly similar architecture to the MLP Baseline with the key difference that the TDT model learns its own text embeddings and utilizes attention between different modalities. We see that the standard error bars are larger for the MLP, and we belive this is the reason for its relatively strong performance on the hard tasks.

Our results are shown in Figure 3. The conclusion from this experiment is two-fold:

1. The causal transformer architecture accessing text and state tokens in a single stream outperforms a concatenation-based approach with pre-encoded text tokens.

2. Increasing the context of the causal transformer leads to stronger performance, indicating that longer sequence modeling can lead to a more powerful model.

### 5.2 HOW EFFECTIVE IS UNSUPERVISED PRETRAINING FOR LEARNING REPRESENTATIONS?

Perhaps the most enticing prospect of TDT is its ability to harness large-scale unlabelled data for pretraining, enabling few-shot behavior observed in language. For our unsupervised pretraining experiments, models are initially pretrained via behavior cloning on a fixed number of unlabelled tasks. These training tasks are sampled from the same corpus as the labelled tasks, but do not include text. For each evaluation task, we sample a small subset of the labelled data, including the evaluation

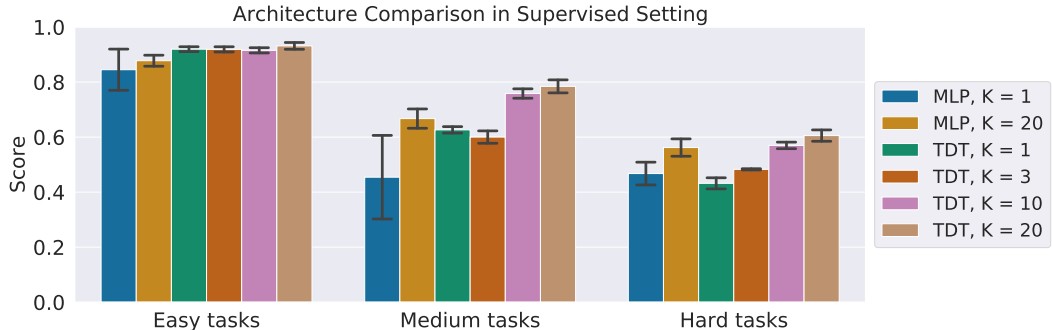

Figure 3: Performance of transformer architecture (TDT), based on the context size $K$. We compare to a concatenation-based MLP architecture which receives encoded text latents from pretrained BERT and concatenates them with the latents of the current state from a learned vision encoder. Text Decision Transformer outperforms the MLP baseline in every suite of tasks. All experiments use 3 seeds and show standard deviation between seeds.

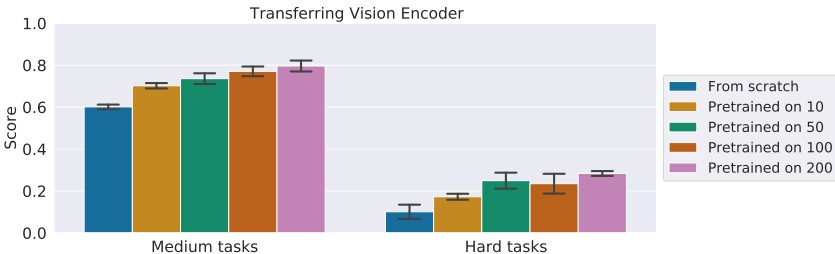

Figure 4: Transferring vision encoder after unsupervised pretraining. After pretraining with behavior cloning on unlabelled trajectories, we see significant benefits on finetuning for medium and hard tasks compared to a randomly initialized vision encoder. Note the context size used is 20.

task itself. For instance, when evaluating on the "jump back and forth between the second and third ice floes" task, the model is trained on this task, as well as some related jumping tasks between the first and second, first and third, etc., ice floes. The data that the model is trained on for each evaluation task is the same across all variants. We finetune all the model parameters.

We first evaluate transferring the vision encoder, which represents the benefit from better image representations. This is the currently the most common method of finetuning RL policies after unsupervised exploration (Yarats et al., 2021), taking advantage of larger datasets for stronger representations, but not transferring learned behaviors. Our results are shown in Figure 4.

We conclude that by pre-training the entire model on an unlabeled dataset of transitions, we learn important features in our visual encoder (despite no text labels). Using this visual encoder in later tasks then allows for better performance when fine-tuning on specific tasks. This improvement scales with larger quantities of pre-training data.

### 5.3    DOES SEQUENTIAL CONTEXT IMPROVE PRETRAINING PERFORMANCE?

Additionally, due to the sequential nature of the transformer, we can further investigate how context length affects pretraining. Since TDT jointly models modalities, there are multiple possible modes of supervision. On the one hand, the vision encoder may learn to pick up on valuable features simply due to repeated actions in specific positions. For instance, whenever the player is located on the bottom ice floe, the agent may tend to jump higher. In such a case, there would be a learning signal from the environment which would allow the encoder to pick-up on features indicating when the agent is in the bottom ice floe. On another hand, the vision representations will be processed sequentially by a transformer, and may learn features which correlate behavior across timesteps. For

instance, if the previous frames indicate an agent jumping between two ice floes, knowledge of this past would allow the model to infer the task at hand.

To evaluate this, we ablate TDT from Figure 4 by using a context size of 1 instead of 20. Again, we transfer the vision encoder only. Our results are shown in Table 2 and Figure 5. We observe that larger contexts do indeed lead to larger percent increases from pre-training. This suggests that the improvement from pre-training is not just due to seeing repeated state-action pairs, and benefits from trajectory-level inference.

| Split | Context = 1 | Context = 20 |
|---|---|---|
| Hard | 43.5% | **181.1%** |
| Medium | 39.0% | 39.0% |

Table 2: Comparing benefit of transferring vision encoder for different model context sizes. Improvement is measured as percent increase in performance when pre-training with 200 unlabelled tasks compared to an untrained model. The improved transfer of the 20 context model indicates that sequential trajectory-level information can be transferred.

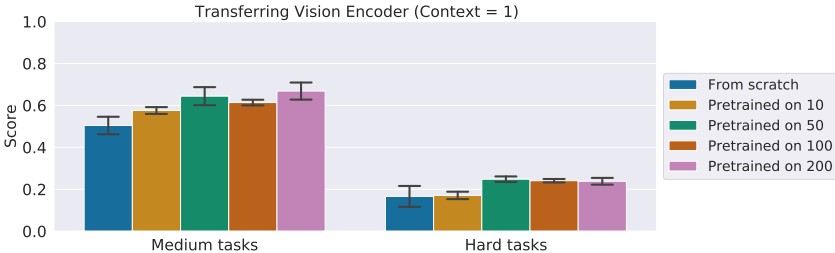

Figure 5: Transferring vision encoder with a shorter context (size 1). While pretraining still helps, it is generally less effective than with the model utilizing a longer context (Figure 4); the numbers for pretraining on 200 unlabelled tasks are summarized in Table 2.

## 5.4 CAN WE TRANSFER THE SEQUENCE MODEL BACKBONE?

In the past sections, we investigated transferring the vision encoder only from the pretraining phase. This has been standard for finetuning RL agents after unsupervised exploration because it can be difficult to transfer the policy/value parameters between the distinct exploration/finetuning regimes. However, as we have seen in language, we can gain greater expressiveness in finetuning by transferring the sequential aspect of the model (Devlin et al., 2018; Radford et al., 2019). We repeat the same unsupervised pretraining phase, but now transfer the transformer backbone, which captures learned behaviors and sequential integration of the inputs.

Our results are shown in Figure 6. The sequence model incorporates additional information that is applicable to the downstream tasks: for instance, after pretraining on 200 unlabelled tasks and then evaluating on the hard tasks, additionally transferring the sequence model doubles the score of only transferring the vision encoder. This hints that transferring learned sequence models can serve as a powerful foundation for few-shot learning in RL settings.

## 5.5 HOW DOES TEXT DECISION TRANSFORMER SCALE WITH DATA?

Finally, we also show additionally scaling results in the supervised setting. In the previous experiments, we showed pretraining on larger amounts of unsupervised data improved performance. Here, we additionally train the model on larger amounts of labelled data ranging from 10 to 700, and then evaluate the performance of these models on a heldout validation set. The validation set consists of new trajectories and some new task statements not in the training data. The validation loss of the model is shown in Figure 7. A low validation loss indicates that the model is more likely to predict the correct action, and hence more likely to achieve higher success rates in the environment.

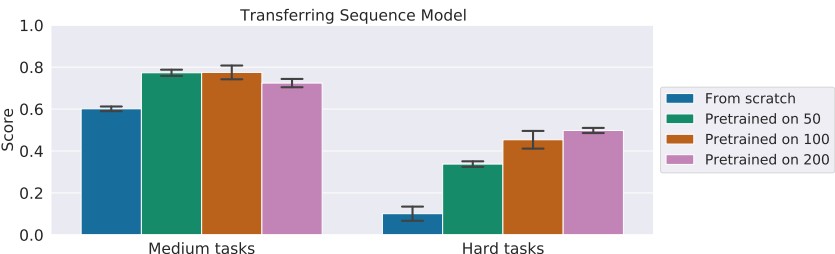

Figure 6: Transferring both the sequence model and the vision encoder after unsupervised pretraining. The sequence model captures additional information that improves performance compared to only transferring the vision encoder.

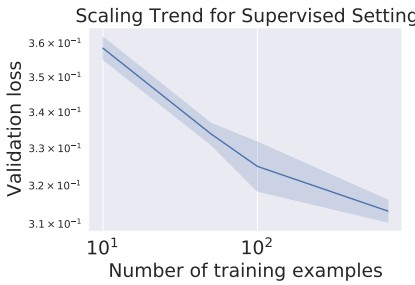

Figure 7: Validation loss scaling in the supervised setting based on number of labelled training trajectories. The relationship between loss and data is approximately linear on a log-log plot.

### 5.6 QUALITATIVE EXAMPLES

See Figure 8 for an example rollout with the task "jump between the second and third ice floes". The agent can be seen jumping between the two aforementioned ice floes.

## 6 RELATED WORK

**Language-conditioned IL/RL.** BabyAI (Chevalier-Boisvert et al., 2019) also proposes a benchmark for evaluating language learning on a griworld task. While BabyAI includes a combinatorially diverse set of tasks using language, the tasks are procedurally generated and thus somewhat less "natural", so the tasks and example trajectories are not as diverse. We propose a more open-ended approach, where the dataset consists of human-generated tasks with no enforced structure of the text, in hopes of training more realistic and capable models. We also use an environment more reliant on skill acquisition, as the model must be able to string together complex action sequences to reach high success rates. CLIPort (Shridhar et al., 2021) studies language for robotic tasks, using CLIP (Radford et al., 2021) to transfer supervision from pure language to images, complementary to our investigation of transferring supervision from pure trajectories to language. Additionally, both BabyAI and CLIPort, like many other works (Chaplot et al., 2018; Stepputtis et al., 2020; Abramson

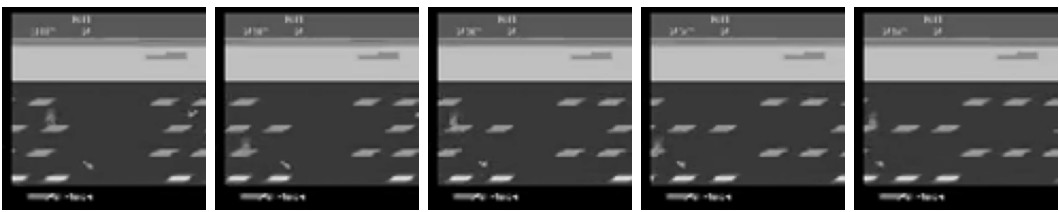

Figure 8: Key frames from a generated agent rollout corresponding to task "jump between the second and third ice floes". States represent a summary of the behavior; see Appendix for longer trajectory.

et al., 2020; Nair et al., 2021), use a multi-stream approach, embedding different inputs/modalities with separate models, in contrast to our single-stream approach which simply extends Decision Transformer (Chen et al., 2021). Our environment differs from CLIPort in that it does not require advanced attention mechanisms to be successful. Indeed, the Atari interface allows for impressive performance even with CNNs and MLPs. Another environment addressing language-conditioned imitation is ALFRED (Shridhar et al., 2020). Unlike ours, this environment provides an overarching task, along with step-by-step instructions to perform the task. Additionally, the time horizon in ALFRED is smaller than in our work (some missions take thousands of steps). Our dataset contains more image-action pairs, and focuses on longer-horizon tasks, but in a simpler environment without sub-tasks. In comparison to other works such as R2R (Anderson et al., 2018) and Touchdown (Chen et al., 2020), our work requires less visual feature engineering due to the simpler input, but retains complex language and allows for longer time horizon missions.

**Transformers.** Our approach to modeling trajectories follows Chen et al. (2021) and Janner et al. (2021), except we condition on behaviors by conditioning on language tokens rather than target returns, which can yield a more diverse set of trajectories. These approaches use sequence modeling, rather than TD learning as other RL transformers works have (Parisotto et al., 2020; Ritter et al., 2020), which may be simpler and more stable. Transformers have also been more broadly applied to multimodal tasks, such as vision and text (Lu et al., 2019; Radford et al., 2021). Like the above discussion, most multimodal approaches use separate models for different models for different modalities, e.g. CLIP (Radford et al., 2021) learns separate models to embed text and images into a similar representation space. Our approach is akin to DALL-E (Ramesh et al., 2021), which uses a single-stream approach to model text and images, where we replaces images with trajectories, and do not predict the text. Prompt tuning (Lester et al., 2021; Li and Liang, 2021) has the potential to turn a single-modality model into a multimodal one with minimal finetuning (Lu et al., 2021; Tsimpoukelli et al., 2021), and would be interesting future work. We showcase positive results from the ability to transfer sequential behavior from unlabeled pretraining, which parallels trends seen in unsupervised language pretraining (Devlin et al., 2018).

# 7 CONCLUSION

We developed a new benchmark, Text-Conditioned Frostbite, and released a corresponding dataset, with the aim of accelerating progress in the development of RL agents which capably utilize language. We also proposed Text Decision Transformer, a multimodal architecture operating over text, state, and action tokens. We showed results showing that unsupervised pretraining can improve downstream few-shot learning, both when transferring the vision encoder and the sequence backbone. We released code and the dataset used, and hope future work will continue exploring approaches for effective few-shot learning in language and RL settings.

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

| Example Tasks | | |
|---|---|---|
| dont finish level 2 | do not go below the first ice floe | hit birds |
| minimize the time spent alive | get to the third level | reach each level as fast as possible |
| start playing once there are 30 seconds left | make sure to collect fish | dont enter the first igloo |
| jump via the second ice floe | complete the first igloo | hit a bird on the third level |
| let the timer run out in every life | run into the left border | fall in the water every life |
| do not pass the second ice floe | dont move | build the first 6 igloos |
| prioritize getting as many fish as possible | dont enter the third igloo | only play on your first life |
| run to the left before starting | dont die from crabs | start playing when there are 20 seconds left |
| build but dont enter the third igloo | hit a bird before entering the first igloo | die from water 4 times |

Table 3: Example tasks in the Text-Conditioned Frostbite dataset. Some of these tasks are not included in evaluation so are not broken into easy, medium, and hard suites.

# A    EXPERIMENTAL DETAILS

In this section, detail performance on a task-by-task basis, and list the tasks which are used for the experiments.

## A.1    TASK SUITES

In the easy task suite, the evaluation tasks are 'dont move', 'stay on the left side', 'try to stay on the right side', 'die on the first level', and 'get as close to 500 points as you can'. In the medium task suite, the evaluation tasks are 'jump back and forth between the second and third ice floes', 'get as close as you can to 1000 points', 'spend as much time as possible on the first ice floe',

## A.2    BASELINES BY TASK

In this section, we include the performances of the baselines by task.

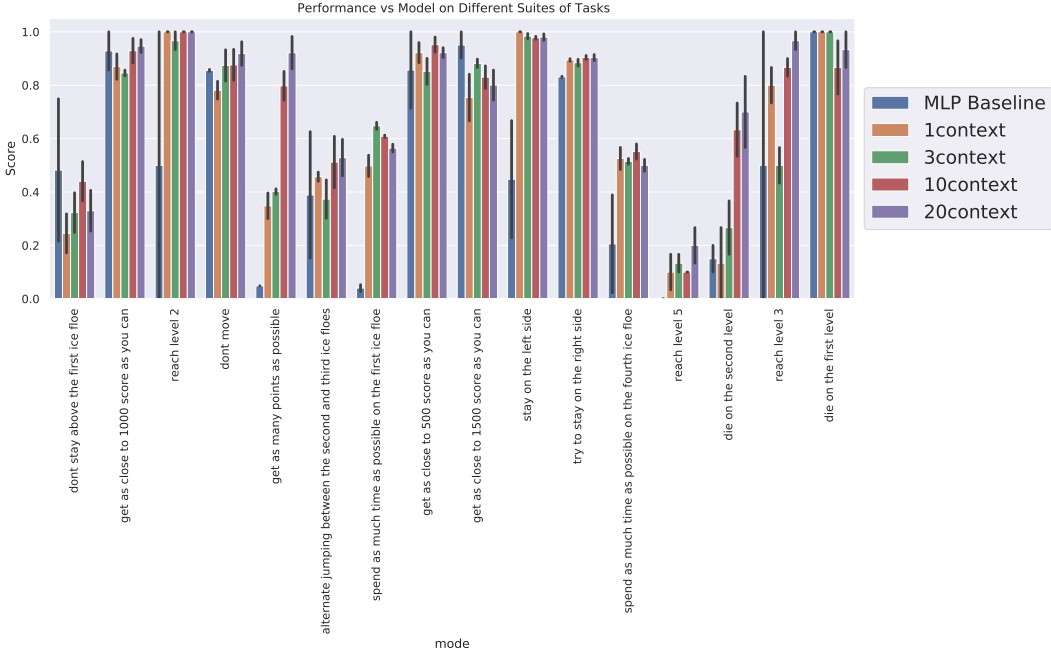

Figure 9: Performance vs. Baseline. Above we see the performance of the model relative to a concatenation-based MLP architecture on a per-task basis.

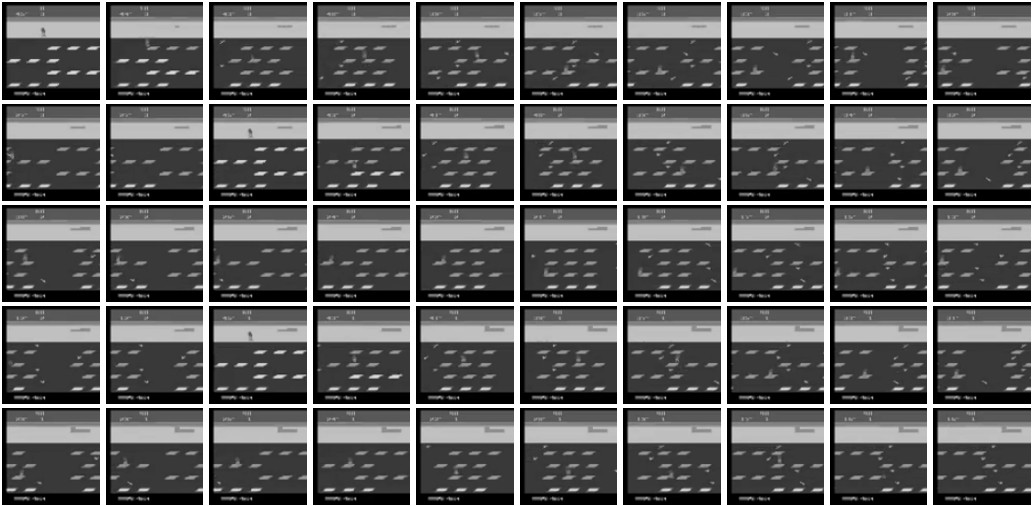

Figure 10: Select frames from a generated agent rollout corresponding to mission "jump between the second and third ice floes". States are sampled from every 50 frames, and visualized left to right, top to bottom. Note that many jumps are still lost by subsampling.

### A.3 FULL TRAJECTORY VISUALIZATION

Please see figure 10 for a more complete sequence of a generated trajectory.

### A.4 HYPERPARAMETERS FOR BASELINE TEXT DT

In this section, we share the hyperparameters we used for the baseline plots with the Text Decision Transformer. Please see table 4

Table 4: Hyperparameters of TDT for Frostbite experiments.

| Hyperparameter | Value |
|---|---|
| Number of layers | 6 |
| Number of attention heads | 8 |
| Embedding dimension | 128 |
| Batch size | 128 |
| Context length $K$ | 20 |
| Text context length | 20 |
| Nonlinearity | ReLU, encoder |
| | GeLU, otherwise |
| Encoder channels | $32, 64, 64$ |
| Encoder filter sizes | $8 \times 8, 4 \times 4, 3 \times 3$ |
| Encoder strides | $4, 2, 1$ |
| Max epochs | 10 |
| Dropout | 0.1 |
| Learning rate | $1 * 10^{-4}$ |
| Adam betas | $(0.9, 0.95)$ |
| Grad norm clip | 1.0 |
| Weight decay | 0.1 |

A.5 ALGORITHM PSEUDOCODE

Here, we provide pseudocode for our implementation.

---

**Algorithm 1** Text-Conditioned Decision Transformer Pseudocode

---

```
# m, s, a, t: tasks, states, actions, or timesteps
# transformer: transformer with causal masking (GPT)
# embed_s, embed_a: linear embedding layers
# embed_m, embed_t: learned discrete embedding tables
# pred_a: linear action prediction layer

# main model
def TextDecisionTransformer(m, s, a, t):
    # compute embeddings for tokens
    m_embedding = embed_m(m)
    s_embedding = embed_s(s) + embed_t(t)
    a_embedding = embed_a(a) + embed_t(t)

    # interleave tokens as (m_1, ..., m_n, s_1, a_1, ..., s_K)
    input_embeds = stack(s_embedding, a_embedding)  # interleave (s, a)
    input_embeds = concatenate(m_embedding, input_embeds)  # prepend text

    # use transformer to get hidden states corresponding to actions
    hidden_states = transformer(input_embeds=input_embeds)
    a_hidden = unstack(hidden_states).actions

    # predict action logits
    return pred_a(a_hidden)

# training loop
for (m, s, a, t) in dataloader:
    a_preds = TextDecisionTransformer(m, s, a, t)
    loss = cross_entropy_loss(a_preds, a)
    optimizer.zero_grad(); loss.backward(); optimizer.step()

# evaluation loop
m, s, a, t, done = [task], [env.reset()], [], [1], False
while not done:  # autoregressive generation/sampling
    # sample next action
    action = TextDecisionTransformer(m, s, a, t).sample()
    new_s, r, done, _ = env.step(action)

    # append new tokens to sequence
    s, a, t = s + [new_s], a + [action], t + [t[-1]+1]
    s, a, t = s[-K:] ...  # only keep context length of K
```

---

