# OpenReview forum: "Pretraining for Language Conditioned Imitation with Transformers"
_ICLR.cc/2022/Conference — ICLR 2022 Submitted_

### Official Review · Reviewer_hbz9 · 2021-10-24

**Correctness:** 3
**Technical Novelty And Significance:** 3
**Empirical Novelty And Significance:** 3
**Recommendation:** 6
**Confidence:** 3

**Main Review:**

Strengths:

1. This paper introduces a quite large text corpus for the text game community.
2. This paper presents some baselines for pretrained text decision transformers.

Weakness:

May need some interpretations for some weird cases in the Table and Figures.

For example,

1). In Table 4, it looks more context does not help for Medium tasks?

2). Similarly, in Figure 3, it looks MLP is quite a strong baseline on the Hard tasks?

**Summary Of The Paper:**

This paper introduces a corpus of text instructions for the language guided Atari Frostbite environment, and also presents a study of pretrained Text Decision Transformer on the corpus, shows that it can improves upon baselines, yield better results in the low-data regimes.

**Summary Of The Review:**

This paper introduces a quite large corpus of text instructions for the language guided Atari Frostbite environment, which is good resources for the community, and it also presents baselines for pretrained text decision transformers (TDT), and shows some interesting empirical results.

---

> ### Author Response · Authors · 2021-11-23
> **Response to Reviewer hbz9**
>
> Thank you for taking the time to review our paper, we hope to address some of your concerns here:
> “In Table 4, it looks more context does not help for Medium tasks?”
>
> We believe this is simply due to fluctuations and variance by seed. We see a strong improvement in performance with context size 10 and 20, and believe that in general more context leads to strong results.
>
> “Similarly, in Figure 3, it looks MLP is quite a strong baseline on the Hard tasks?”
>
> We believe this is due to the larger variance in the model, and that with small context the models struggle. In our updated plot, we include more context and see that the TDT models have a significant benefit over the MLP.

---

### Official Review · Reviewer_ZRZY · 2021-11-01

**Correctness:** 2
**Technical Novelty And Significance:** 1
**Empirical Novelty And Significance:** 1
**Recommendation:** 3
**Confidence:** 4

**Main Review:**

I think this paper lacks important detail in that it proposes a new benchmark but declines to compare such benchmark to existing work. The claim that "transferring sequential knowledge is relatively unstudied" is just simply not true. There has been several works, old and recent, on instruction following. These include the large quantity of visual instruction following tasks such as R2R (https://bringmeaspoon.org/), Room-Across-Room (https://ai.google.com/research/rxr/), Touchdown (https://arxiv.org/abs/1811.12354), DRIF (https://github.com/lil-lab/drif), ALFRED (http://askforalfred.com/), as well as complex grid worlds with new language-entity associations and hard generalization requirements (https://arxiv.org/abs/1910.08210, https://arxiv.org/abs/2101.07393). In order to propose this benchmark, the paper ought to compare the proposed environment with these existing work - why is there a need for this new benchmark? In what ways is it different than previous environments? This crucial question is not addressed in this work. These related work and many others (https://arxiv.org/abs/1906.03926) are not even cited. Another point of contention I have with the way this work is posed is that it claims 5 million annotated transitions. What does this mean? It seems like there are only a few hundred labeled trajectories. Do the authors means they have a couple hundred labeled trajectories, each with 10k steps (where the task is fixed)? If so, I think this is at best misleading - no other dataset for language grounding that I am aware of lists annotation size like this.

Here are some concrete examples of claims with missing analyses:
1. "a diverse set of tasks" - how are these tasks diverse? What do the tasks even look like? The authors mention 500 tasks but there are only  8 listed in the appendix. Upon inspection of these tasks, they seem very unusual (die on the x level, try to get as close as you can to 1000 score). How many unique tasks are there? What does it even mean to "get as close as you can"? Or "spend as much time as possible"? How is the reward assigned?
2. "temporally extended strategies" - what does this mean? what is the distribution of trajectory lengths of annotated playthroughs per task? of agent playthroughs per task?

Moreover, what is the evaluation set? How is this set constructed? What skills is it evaluating? Is generalization required? If an agent just cycles through memorized action patterns from the training distribution, can it still do well on the test set?

**Summary Of The Paper:**

This work proposes a new instruction-following reinforcement learning environment based on the Atari Frostbite environment. They experiment with a modified decision transformer on the environment. The results largely show that performance improves with longer context, more pretraining, and more data.

**Summary Of The Review:**

No comparison to prior tasks, unclear what new proposed environment tests and how it compares to prior environments.

---

> ### Author Response · Authors · 2021-11-23
> **Response to Reviewer ZRZY**
>
> Thank you for taking the time to review our paper. We appreciate your feedback and hope to have addressed some of your concerns here:
> Thank you for taking the time to review our paper. We hope we have addressed some of your concerns in our general response regarding novelty and the motivation behind designing this benchmark.
>
> “The claim that "transferring sequential knowledge is relatively unstudied" is just simply not true. There has been several works, old and recent, on instruction following.”
>
> Thanks for your point, we reword this to sequential behavior knowledge. Our goal is to transfer specifically behavioral knowledge, without access to text labels during pre-training.
>
> “In what ways is it different than previous environments?”
>
> We believe the importance of our environment/baselines is due to a few factors. First, the trajectories and trajectory labels are generated by hand. This is in contrast to BabyAI,  where text and labelled trajectories are programmatically generated, and thus contain fewer variations in natural language and action/ strategy choice; in some sense these languages are "synthetic". In these benchmarks, we might think of the models as learning essentially one-hot object representations, rather than an understanding of "natural" language. Second, the environment is part of the Atari suite. Atari games are well-studied in the context of reinforcement learning and there is well-established community knowledge regarding standards for training in the environment (vision encoders, etc.). This allows for a more level playing field for comparing models and less engineering is required. This is in contrast to RAVENs (CLIPort) where feature engineering is much more complex (transporter network outperforms all). Lastly, the Atari Frostbite environment requires temporally extended strategies. Certain missions such as reaching a specific level require thousands of timesteps to be successfully completed. This in turn requires more advanced planning / control of the dynamics of the environment. This is as opposed to environments like ALFRED, DRIF, R2R, which have shorter time horizons (and some have submissions, which further simplify the objective).
>
>
> “Another point of contention I have with the way this work is posed is that it claims 5 million annotated transitions. What does this mean?”
>
> Regarding your question about the number of annotations, we apologize if it comes off as misleading. We have re-done the wording to clarify that it is indeed several hundred missions, each with approximately 10k frames of transitions. Thank you for pointing out that it perhaps misrepresented the dataset.
>
> “"a diverse set of tasks" - how are these tasks diverse? The authors mention 500 tasks but there are only 8 listed in the appendix. Upon inspection of these tasks, they seem very unusual”
>
> We think your discussion about clarifying the meaning of the tasks is very valuable. Indeed, some of the tasks may have slightly ambiguous meanings, ones that may not intuitively be easily solvable by traditional learning methods. We used phrases like “spend as much time as possible” to capture phrases that humans would likely be able to solve, but still lack the crystal clarity of other benchmarks. We have included more task examples in the appendix to help paint a clearer picture of the training dataset.
>
> “How is the reward assigned?”
>
> Reward is assigned in a similar fashion to DM-control: we use the underlying state-space of the game to calculate specific metrics in the environment. These metrics are then normalized to be in the range (0, 1), so no task gets unequal weighting. For example, in a task like “spend as much time as possible on the first ice floe”, the assigned success is simply (# timesteps on 1st ice floe) / (# timesteps).
>
> “Moreover, what is the evaluation set? How is this set constructed?”
>
> The evaluation set is a set of specific tasks and their corresponding reward functions. The set of tasks is pre-determined, and is held constant across all models to ensure fairness. These tasks evaluate how well the model is able to perform a specific task, with reward coming from the underlying state space of the game.

---

### Official Review · Reviewer_ZbHJ · 2021-11-01

**Correctness:** 3
**Technical Novelty And Significance:** 2
**Empirical Novelty And Significance:** 2
**Recommendation:** 3
**Confidence:** 4

**Main Review:**

Pros
* Learning behavior policies from language supervision is an important problem setting
* Useful to study the applicability of the transformer architecture to sequential decision making problems
* New benchmark on language conditioned agent learning

Cons
* The new proposed benchmark needs to be contextualized in existing work. Why is there a need for this benchmark and how is it different from existing language instruction following benchmarks from the embodied learning literature (Eg. VLN benchmarks like Room to room and object interaction benchmarks like ALFRED)?
* Models and training approaches considered in the paper seem straightforward. What are the new technical contributions of the work?
* Weak experiments

I would encourage the authors to think about what’s new and interesting about the proposed new dataset, what makes it challenging and how it is related to and different from other existing benchmarks.

Problem setting is unclear - In section 5.2, I don’t understand why the model is trained on the evaluation task. Also, the section ends abruptly without a conclusion.

The conclusions on pretraining and comparisons against context size 1 seem obvious and expected. I didn't learn much from these experiments. The experiments needs to provide more insight into the dataset and the results.

Presentation
- Alg 1 can be moved to the appendix.
- The MLP baseline wasn’t clearly described.

**Summary Of The Paper:**

This paper proposes a new language conditioned imitation learning task based on the Atari Frostbite environment. In addition, the paper studies imitation learning with a transformer architecture and the effect of pre-training on game trajectories.

**Summary Of The Review:**

The paper lacks novel technical contributions. The experiments are weak and do not provide new, meaningful and practically useful insights. The work further needs to be contextualized better in existing work.

---

> ### Author Response · Authors · 2021-11-23
> **Response to Reviewer ZbHJ**
>
> Thank you for taking the time to review our paper. We appreciate your feedback and hope to have addressed some of your concerns here:
> “The new proposed benchmark needs to be contextualized in existing work. Why is there a need for this benchmark and how is it different from existing language instruction following benchmarks from the embodied learning literature (Eg. VLN benchmarks like Room to room and object interaction benchmarks like ALFRED)?”
>
> We believe the importance of our environment/baselines is due to a few factors. First, the trajectories and trajectory labels are generated by hand. This is in contrast to BabyAI,  where text and labelled trajectories are programmatically generated, and thus contain fewer variations in natural language and action/ strategy choice; in some sense these languages are "synthetic". In these benchmarks, we might think of the models as learning essentially one-hot object representations, rather than an understanding of "natural" language. Second, the environment is part of the Atari suite. Atari games are well-studied in the context of reinforcement learning and there is well-established community knowledge regarding standards for training in the environment (vision encoders, etc.). This allows for a more level playing field for comparing models and less engineering is required. This is in contrast to RAVENs (CLIPort) where feature engineering is much more complex (transporter network outperforms all). Lastly, the Atari Frostbite environment requires temporally extended strategies. Certain missions such as reaching a specific level require thousands of timesteps to be successfully completed. This in turn requires more advanced planning / control of the dynamics of the environment. This is as opposed to environments like ALFRED, DRIF, R2R, which have shorter time horizons (and some have submissions, which further simplify the objective).
>
> “Models and training approaches considered in the paper seem straightforward. What are the new technical contributions of the work?”
>
> While the alterations to previous architectures are small, our key focus is the emergence of transfer of sequential behavior -- which is particularly interesting in the long-horizon Atari Frostbite setting -- rather than novelty in architecture (ours is close to GPT and Decision Transformer) or transferring the vision encoder or pretraining with langauge. However, we hope that the relatively simple tweaks made to these models encourage understanding and wide-spread use of our models. The formulation of the pre-training procedure again is quite straightforward, but we believe it is nevertheless an important contribution to display our performance gains, and can build the foundation for RL models which learn generally transferable skills in an end-to-end fashion, like language models.
>
> “The conclusions on pretraining and comparisons against context size 1 seem obvious and expected. I didn't learn much from these experiments. The experiments needs to provide more insight into the dataset and the results.”
>
> We have updated the plot to include a baseline that uses more context. We have altered our analysis and hope this provides more insight into the benefits of the attention-based architecture.
>
> “The conclusions on pretraining and comparisons against context size 1 seem obvious and expected.”
>
> We have added a baseline with more context, to provide more insight to the benefits of the architecture.
>
>
> Thank you for taking the time to review our paper. We appreciate your clarifying questions regarding the main contributions in the work, and hope to have addressed them in our general response. We have added more analysis in the paper for the experiments you have mentioned.

---

> > ### Comment · Reviewer_ZbHJ · 2021-11-29
> > **Rebuttal response**
> >
> > I thank the authors for the response. I continue to feel that the work has limited technical novelty and needs to better justify the need for the proposed benchmark. I would like to keep my score.

---

### Official Review · Reviewer_EZAA · 2021-11-03

**Correctness:** 3
**Technical Novelty And Significance:** 2
**Empirical Novelty And Significance:** 2
**Recommendation:** 5
**Confidence:** 4

**Main Review:**

I agree that this paper addresses an important problem of language-conditioned imitation learning. However, It seems that the contribution of the proposed model, Text Decision Transformer (Text-DT), is slightly marginal and Text-DT seems not related to Decision Transformer. The detailed comments and questions are as follows:

1. The biggest difference between standard Transformer and Decision Transformer is the return-to-go, but there is no return-to-go in Text-DT. The Text-DT, proposed in this paper, is thought to be just using the standard transformer as a conditional sequence generation method, so I am not sure if it is right to call it a decision transformer. Why is it called a (Text) decision transformer?

2. The authors use Text-DT and TDT interchangeably in their papers, is there any difference? If there is a difference, it should be clearly explained, and if there is no difference, I think the terminology should be unified.

3. (Figure 3) In easy/medium tasks, TDT(k=1) performs better than MLP baseline, but in hard tasks, MLP baseline shows better performance than TDT(k=1). What is the reason for this?

4. As an experiment in Figure 3, how does the performance of the MLP baseline change as the context length increases? It would be better if the authors could compare and show them together.

5. The authors describe the proposed Text-DT model as the main contribution, but there does not seem to be much difference from the existing works using the standard transformer architecture. Rather, it would be better to show the contribution to the point of proposing a new benchmark more intensively.



**Summary Of The Paper:**

This paper presents a new multimodal benchmark for language conditioned RL settings, where an agent must complete tasks specified by text instructions in the Atari Frostbite environment. Their benchmark provides a dataset of 5M text-labeled transitions for training. Finally, the authors propose a model for language conditioned RL settings based on Transformer architecture as a baseline.

**Summary Of The Review:**

I think that this paper presents an interesting benchmark for developing RL agents which can utilize language. However, the proposed model, Text-DT, is considered to have a small contribution because the Transformer architecture is simply applied to the conditional sequence generation problem with language inputs. In addition, the authors explain the proposed model based on the decision transformer, but it is not considered to have much related to the decision transformer. I would like to recommend a more detailed explanation of the proposed benchmark and possible future studies.

---

> ### Author Response · Authors · 2021-11-23
> **Response to Reviewer EZAA**
>
> Thank you for taking the time to review our paper. We appreciate your feedback, and hope to have addressed some of your concerns:
> “The biggest difference between standard Transformer and Decision Transformer is the return-to-go, but there is no return-to-go in Text-DT… Why is it called a (Text) decision transformer?”
>
> We have adopted the name Text Decision Transformer to clarify that the architecture is related to the original decision transformer, namely that we care about a sequential context (namely, the previous states) and are making decisions using it. While the original DT focused on returns-to-go for policy conditioning, we also heavily utilize the past states (and text); in particular, the sequence modeling aspect of the past states and actions can be transferred from a pretraining phase.
>
> “The authors use Text-DT and TDT interchangeably in their papers, is there any difference? If there is a difference, it should be clearly explained, and if there is no difference, I think the terminology should be unified.”
>
> There is no difference between these models, and we have clarified this in the paper.
>
> “(Figure 3) In easy/medium tasks, TDT(k=1) performs better than MLP baseline, but in hard tasks, MLP baseline shows better performance than TDT(k=1). What is the reason for this?”
>
> We believe this is simply due to the larger variance in results for the MLP models that we trained.
>
> “​​As an experiment in Figure 3, how does the performance of the MLP baseline change as the context length increases? It would be better if the authors could compare and show them together.”
>
> Thank you for suggesting to increase the context size of the MLP baseline. We have added this model, and include the plot in the paper. The MLP baseline performance increases, but still does not match the TDT architecture with equal context.
>
> “The authors describe the proposed Text-DT model as the main contribution, but there does not seem to be much difference from the existing works using the standard transformer architecture.”
>
> While the alterations to previous architectures are small, our key focus is the emergence of transfer of sequential behavior -- which is particularly interesting in the long-horizon Atari Frostbite setting -- rather than novelty in architecture (ours is close to GPT and Decision Transformer) or transferring the vision encoder or pretraining with langauge. However, we hope that the relatively simple tweaks made to these models encourage understanding and wide-spread use of our models. The formulation of the pre-training procedure again is quite straightforward, but we believe it is nevertheless an important contribution to display our performance gains, and can build the foundation for RL models which learn generally transferable skills in an end-to-end fashion, like language models.

---

> > ### Comment · Reviewer_EZAA · 2021-11-29
> > **Response to rebuttal**
> >
> > Thank you for taking the time to respond to my questions. I have read the other reviewer's review and the author's overall response. However, I still feel that the proposed method is quite straightforward and the technical contribution of this paper is small. So, I will keep my score.

---

### Author Response · Authors · 2021-11-23
**General Response**

First, we wanted to thank the reviewers for taking the time to read through our paper. We appreciate the insightful comments, suggestions, and clarifying questions. Here we’ll discuss some of the common concerns raised by reviewers ZbHj and ZRZY: The first recurring concern is a lack of justification for our benchmark. We believe the importance of our environment/baselines is due to a few factors. First, the trajectories and trajectory labels are generated by hand. This is in contrast to BabyAI,  where text and labelled trajectories are programmatically generated, and thus contain fewer variations in natural language and action/ strategy choice; in some sense these languages are "synthetic". In these benchmarks, we might think of the models as learning essentially one-hot object representations, rather than an understanding of "natural" language. Second, the environment is part of the Atari suite. Atari games are well-studied in the context of reinforcement learning and there is well-established community knowledge regarding standards for training in the environment (vision encoders, etc.). This allows for a more level playing field for comparing models and less engineering is required. This is in contrast to RAVENs (CLIPort) where feature engineering is much more complex (transporter network outperforms all). Lastly, the Atari Frostbite environment requires temporally extended strategies. Certain missions such as reaching a specific level require thousands of timesteps to be successfully completed. This in turn requires more advanced planning / control of the dynamics of the environment. This is as opposed to environments like ALFRED, DRIF, R2R, which have shorter time horizons (and some have sub-missions, which further simplify the objective).

Second, we hoped to address some of your concerns regarding the technical contributions of the model. While the alterations to previous architectures are small, our key focus is the emergence of transfer of sequential behavior -- which is particularly interesting in the long-horizon Atari Frostbite setting -- rather than novelty in architecture (ours is close to GPT and Decision Transformer) or transferring the vision encoder or pretraining with language. However, we hope that the relatively simple tweaks made to these models encourage understanding and wide-spread use of our models. The formulation of the pre-training procedure again is quite straightforward, but we believe it is nevertheless an important contribution to display our performance gains, and can build the foundation for RL models which learn generally transferable skills in an end-to-end fashion, like language models.

---

### Decision · Program_Chairs · 2022-01-20

**Decision:**

Reject

**Comment:**

The arguments the paper makes require a stronger foundation and justification. The reviewers and AC didn't find the author response sufficient. For example, in response to ZbHJ, the authors argue that their benchmark doesn't use automatically generated trajectories and therefore the language is not synthetic in some sense. It's not clear how it's related to synthetic language, but generated trajectories does create artificial regularities in the task, so an issue, but one that the author must address accurately. This argument also seems to focus on ALFRED and R2R, and ignored many other benchmarks, like the data used in DRIF (mentioned later), RxR, Touchdown, etc. There is also mis-used of technical term (e.g., Decision Transformer). Generally, the reviewers consider the work of potential, but it requires significant refinement, which the author response did not provide.